# Intra- and inter-operator reliability of measuring compressive stiffness of the patellar tendon in volleyball players using a handheld digital palpation device

Lotte van Dam[1,2,3]*, Rieneke Terink[1], Inge van den Akker-Scheek[4], Johannes Zwerver[1,3]

1 Department of Sports Medicine, Sports Valley, Gelderse Vallei Hospital, Ede, The Netherlands,
2 Department of Human Nutrition and Health, Wageningen University, Wageningen, The Netherlands,
3 Center for Human Movement Sciences, University of Groningen, University Medical Center Groningen, Groningen, The Netherlands, 4 Department of Orthopedics, University of Groningen, University Medical Center Groningen, Groningen, The Netherlands

* lottevandam@zgv.nl

**Data Availability Statement:** All relevant data are within the paper and its Supporting Information files. However, the variables length and BMI have

## Abstract

This observational study aimed to evaluate the intra- and inter-operator reliability of a digital palpation device in measuring compressive stiffness of the patellar tendon at different knee angles in talent and elite volleyball players. Second aim was to examine differences in reliability when measuring at different knee angles, between dominant and non-dominant knees, between sexes, and with age. Two operators measured stiffness at the midpoint of the patellar tendon in 45 Dutch volleyball players at 0˚, 45˚ and 90˚ knee flexion, on both the dominant and non-dominant side. We found excellent intra-operator reliability (ICC>0.979). For inter-operator reliability, significant differences were found in stiffness measured between operators (p<0.007). The coefficient of variance significantly decreased with increasing knee flexion (2.27% at 0˚, 1.65% at 45˚ and 1.20% at 90˚, p<0.001). In conclusion, the device appeared to be reliable when measuring compressive stiffness of the patellar tendon in elite volleyball players, especially at 90˚ knee flexion. Inter-operator reliability appeared to be questionable. More standardized positioning and measurement protocols seem necessary.

## Introduction

Patellar tendinopathy is an injury of the patellar tendon with a high prevalence among athletes participating in sports that require repetitive jumping and cutting maneuvers. In basketball and volleyball 32% to 45% of elite players have experienced patellar tendinopathy [1]. Patellar tendinopathy can result in sustained or repetitive symptoms, [2] and influences not only sports performance but also daily and work activities [3]. Almost 60% of patients with patellar tendinopathy experience problems performing physically demanding work [4]. Load management, pain education and progressive tendon loading exercises are the recommended treatment. However, treatment is not always successful [3,5–8].

been removed in order to protect the athletes' identities.

**Funding:** This project was funded by a SportInnovator Voucher from ZonMw (file no.: 538001779). The funders had no role in study design, data collection and analysis, decision to publish, or preparation of the manuscript.

**Competing interests:** All authors affirm that they have no financial affiliation (including research funding) or involvement with any commercial organization that has a direct financial interest in any matter included in this manuscript.

Alterations in stiffness of tendon tissue have been reported because of training [9–11] as well as in pathological tendons [12–15]. Monitoring stiffness and adjusting tendon load when changes in stiffness occur might be helpful to prevent patellar tendinopathy in athletes. Currently, ultrasound-based imaging techniques including elastography are used to assess stiffness of tendon tissues. However, these measurements require expensive ultrasound devices and expertise, which makes them less practicable for daily use in a sports environment.

Measuring compressive stiffness of the patellar tendon with a handheld digital palpation device, the Myoton, might be a more viable alternative for on-site measurements, as it is portable and easy to use [16]. This stiffness-measuring method does not require expensive equipment or highly skilled practitioners, and it can be performed everywhere. The Myoton, which measures tissue stiffness with a brief pulse to the skin and underlying muscle, fat, or tendon tissue, appeared to be valid and reliable in measuring viscoelastic muscle properties [17,18]. It also seems promising in measuring Achilles and patellar tendon (PT) stiffness [16,19–24]. We found only two articles investigating the reliability of the Myoton, both with methodological limitations, such as statistical tests performed [20,24]. This study aimed to evaluate intra- and inter-operator reliability of a handheld digital palpation device in measuring compressive stiffness of the patellar tendon at different knee angles in talent and elite volleyball players. Second aim was to examine differences in reliability when measuring at different knee angles, between dominant and non-dominant knees, between sexes, and with age.

## Materials and methods

### Study setting

An observational single-center study was conducted at the Dutch Olympic Sports Center Papendal. All participants were recruited and all stiffness measurements were performed in the medical room next to the training location of the national volleyball teams during a regular training period between 30 May 2022 and 14 June 2022, without matches between measurements.

### Ethics

The study was conducted according to the principles of the Declaration of Helsinki (64th WMA Assembly, October 2013). Only observational coded data were used. This study received a non-WMO declaration from the Medical Ethics Committee Oost-NL on 21 February 2022 (Approval No. 2022(13499)), as the Dutch law on Research Involving Human Subjects Act (WMO) did not apply.

### Participants

Participants were recruited via the network of volleyball coaches at the Olympic Center, and all participants who were interested received an information letter about the study protocol. Male and female volleyball players older than 16, who performed strength training as part of their normal training routine were eligible. Participants were excluded if they had any current musculoskeletal dysfunction or took medication that could affect musculoskeletal function. All participants provided written informed consent before participation.

### Tendotonometry

All stiffness measurements were performed with the MyotonPRO (device code 1308600502, SN000041, Tallinn, Estonia). The Myoton is a non-invasive handheld digital palpation device for compressive and measuring of muscle, tendon, and other soft tissue properties. It was

developed to measure muscle elastic properties, hence the name "Myoton". In this project we measured tendon elastic properties, therefore introducing the term "tendotonometry" for the tendon stiffness measuring method with these types of devices. The Myoton applies a brief pulse to the skin overlying the tendon; thereafter several oscillation parameters are used to calculate the mechanical properties of the tissues, including stiffness [12,17]. A trained operator kept the Myoton in one hand, stabilizing that hand with the other hand. The probe of the device was held perpendicular to the skin overlying the PT (±5˚, monitored by the device itself), with a deformation area of 7.1mm$^2$. Next, the probe was pushed against the skin to reach the correct depth. This was signaled by a red light turning green on the Myoton, indicating a pre-compression strength of 0.18N. Then, five short impulses of 0.4N and with a tap interval of 0.8sec, were automatically applied by the device, to induce mechanical, damped oscillations in the underlying tissues. The Myoton provides mean values on the dynamic stiffness (S, N/m) of the five impulses delivered. This stiffness value is calculated by the device with the maximum acceleration of the oscillation and the deformation of the tissue detected by the transducer.

## Study procedure

After giving informed consent, participants filled out a short questionnaire about demographic and training and performance characteristics. Thereafter stiffness measurements were performed. Stiffness was measured halfway between the patellar apex and the tibial tuberosity. This location was chosen for standardization and uniformity.

## Intra-operator reliability

For all participants, the midpoint of the PT was marked by one trained operator on all test days and the location remained visible throughout all measurements. Measurements were performed on both knees, at 0˚, 45˚ and 90˚ knee flexion (Fig 1). Multiple knee angles were chosen to investigate the influence of knee positioning on reliability. A calibrated and validated digital goniometer (Goniometer Pro, Android version, Digiflex Labs, Seattle, USA) [25] was used to quantify the angle of knee flexion. For the 90˚ knee angle, the participant sat in upright position with feet hovering off the floor. At 0˚ and 45˚ knee flexion, the participant was in supine position with legs extended (0˚) or passively half-flexed (45˚) and supported by the examination table (Fig 1). Participants were instructed to lie fully relaxed. The order of measurements was: first the left knee, followed by the right knee, and each knee first at 90˚, followed by 0˚ and 45˚ knee flexion. Three repeated measurements were performed at each of the three different knee angles, on both knees, by one operator (Table 1). One operator performed each measurement without looking at the outcome, then handed over the Myoton to another researcher who wrote down the outcome and handed back the Myoton for the next measurement.

## Inter-operator reliability

Two different operators measured stiffness to determine inter-operator reliability of the Myoton. Each operator measured stiffness 15 times per participant, in five participants (Table 1). These measurements were performed immediately after the abovementioned measurements. Time interval between operator 1 and operator 2 was maximum 10 minutes. Both operators defined the midpoint of the PT by themselves and marked it on the skin. Between operators, the marker on the skin was removed. The location of the midpoint of the PT had to be located again by the second operator. Participants were seated with their knees flexed in 90˚ and feet not touching the floor (Fig 1A). Stiffness was measured at 90˚ flexion and on the dominant

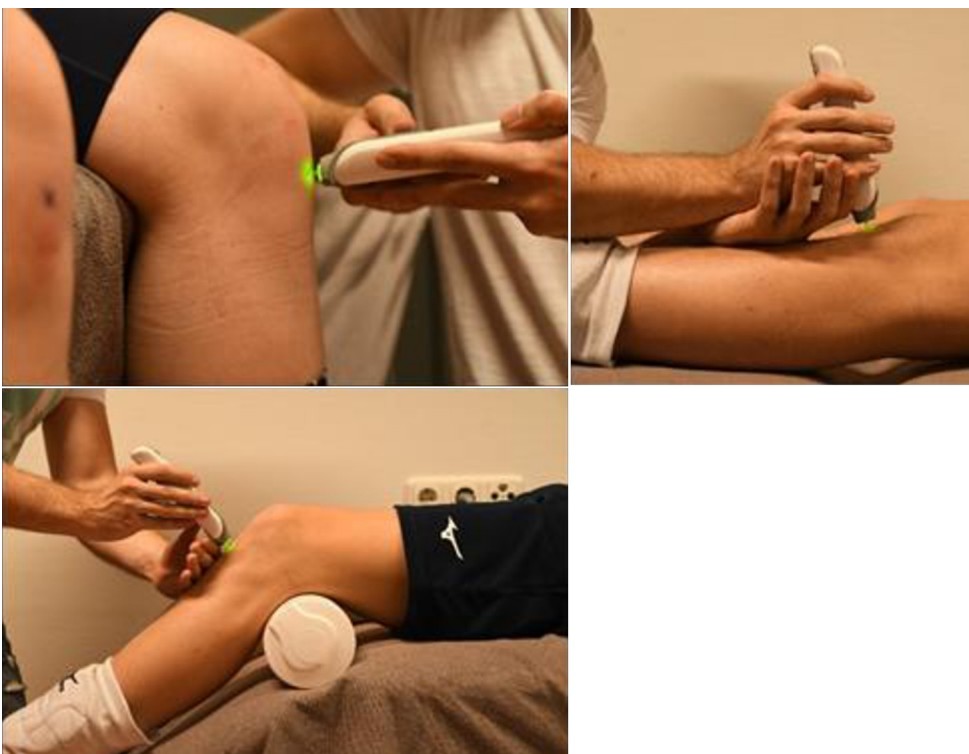

**Fig 1. The three different positions in which stiffness was measured.** All measurements were done in relaxed position and the location on the tendon for measurements was marked on the skin.

knee only. The two operators performed each measurement without looking at the outcome, then handed over the Myoton to another researcher who wrote down the outcome and handed back the Myoton for the next measurement. In that way, both operators were blinded to their own and each other's findings.

## Statistical analysis

Results were analyzed using IBM SPSS Statistics 24 (Armonk, New York, USA). Participants' demographic characteristics were assessed by descriptive statistics. Normality was checked with the Shapiro-Wilk test and frequency histograms. As data were normally distributed, they were presented as mean ± SD or presented graphically in Microsoft Excel 2016. Statistical significance was set at $p < 0.05$.

For baseline characteristics, to investigate differences between sexes, an independent t-test was used for non-categorical data, and a chi-square test for categorical data. To determine differences in stiffness between knee angles, we performed a one-way ANOVA with a Tukey post-hoc test.

**Table 1. Overview of measurements performed to determine reliability of the Myoton.**

|  | Intra-operator reliability | Inter-operator reliability |
|---|---|---|
| No. operators | 1 | 2 |
| No. participants | 45 | 5 |
| No. knee angles | 3 (0˚, 45˚, 90˚) | 1 (90˚) |
| No. measurements per knee angle | 3 | 15 |

For the first aim of this study, we determined reliability of the Myoton in several ways. Intraclass correlation coefficient (ICC) was used to assess intra-operator (2,1, two-way random model, single measures) and inter-operator reliability (2,2, two-way mixed model, mean measures). Reliability was considered excellent when ICC values exceeded 0.75, good-to-fair between 0.40 and 0.75, and poor below 0.40 [26]. Coefficient of variation (CV) was calculated for all measurements (both intr-a and inter operator reliability). Maximum CV allowed was preset at 3% [27]. Amount and percentage of measurements above the 3% border were calculated. Measurement error was determined by calculating general standard error of measurement (SEM; SEM = Standard Deviation, SD * SQRT(1-ICC)), minimal detectable change (MDC; MDC = 1.96 * SEM * SQRT(2)), and 95% limits of agreement (LOA). Bland-Altman plots were made to visualize degree of agreement and to identify systemic bias. In the Bland-Altman plots, LOA and 95% confidence interval (CI) of the mean difference (mean$_{diff}$ ± t$_{n-1, a0.05}$ * (SD$_{diff}$/sqrt(n))) were included. A paired samples t-test was performed to compare stiffness measured by the two operators.

For the second aim of the study, statistical analyses were performed to determine reliability of the Myoton in different situations. A one-way ANOVA was performed to compare the CVs of stiffness between the three knee angles, followed by a post-hoc analysis with Tukey HSD adjustment. Independent t-tests were performed to compare the CVs of stiffness between the dominant vs non-dominant leg and between sexes. A Pearson correlation coefficient was calculated to investigate links between the calculated CVs and age.

## Results

### Basic characteristics

Forty-five healthy volleyball players (16 females, 29 males) participated in this study (Table 2). Stiffness was significantly higher in males than females, except for the 0° knee flexion in the non-dominant knee (S1 Table). In addition, stiffness significantly increased with increasing knee angle (S1 Table, with all post-hoc tests p-value ≤0.006).

### Intra-operator reliability

All ICC values including the confidence intervals were above 0.970 (Table 3), which means excellent intra-operator reliability. In all cases, the average CV was below the 3% border (Table 4).

**Table 2. Summary of basic characteristics of all participants, and differences between males and females.**

| Mean ± SD | All (n = 45) | Females (n = 16) | Males (n = 29) | p-value |
|---|---|---|---|---|
| Age (years) | 17.4±1.1 | 17.0±0.7 | 17.7±1.3 | 0.025 |
| Height (cm) | 193±10 | 185±7 | 198±7 | <0.001 |
| Weight (kg) | 81.1±11.1 | 72.7±7.7 | 85.8±10.0 | <0.001 |
| Body mass index (kg/m$^2$) | 21.6±1.7 | 21.3±1.4 | 21.8±1.9 | 0.331 |
| Dominant leg, Left/Right | 41 L to 4 R | 15 L to 1 R | 26 L to 3 R | 0.644 |
| Training hours per week | 17.9±4.9 | 19.9±3.9 | 16.8±5.1 | 0.029 |
| Strength training per week (h) | 3.6±1.1 | 4.2±0.8 | 3.3±1.1 | 0.004 |
| Dominant knee stiffness (N/m) | 834.6±153.8 | 745.4±151.1 | 883.8±133.7 | 0.005 |
| Non-dominant knee stiffness (N/m) | 833.3±125.3 | 764.2±124.4 | 871.4±110.2 | 0.008 |

Data were obtained using a short questionnaire. Stiffness presented of the patellar tendon, placed in 90° knee flexion.

**Table 3. ICC values of the threefold measured stiffness.**

| Knee | Angle | ICC (95% CI) |
|------|-------|--------------|
| Dominant | 0˚ | 0.988 (0.980–0.993) |
| | 45˚ | 0.985 (0.974–0.991) |
| | 90˚ | 0.991 (0.986–0.995) |
| Non-dominant | 0˚ | 0.986 (0.978–0.992) |
| | 45˚ | 0.985 (0.976–0.991) |
| | 90˚ | 0.987 (0.978–0.992) |

Measured by one operator in 45 volleyball players.

SEM ranged from 11.0 to 15.4 N/m and the MDC ranged from 30.4 to 42.8 N/m (Table 5). Bland-Altman plots showed little to no systemic bias between measurements (S1 Fig). For all cases, zero lay within the 95% LOA and the 95% CI of the mean difference.

## Inter-operator reliability

ICC values showed excellent reliability between two operators (ICC 0.898 (95% CI: 0.833–0.937). However, the measured stiffness differed significantly between operators for all five participants (for p-values, see Table 6). Operator 2 seemed to have more variation compared to operator 1 (3 out of 5 measured participants had a CV above the 3% border, whereas operator 1 had 1 out of 5 measured participants with a CV above the 3% border (Table 6)), and also had an average CV above the 3% border.

SEM for inter-operator measurements was 61.2 N/m, MDC was 169.5 N/m. The 95% LOA for inter-operator stiffness measurements was -195.7–138.2 and the 95%CI of the mean difference was -134.5–77.0.

## Factors influencing reliability

The average CV per knee angle decreased significantly with increasing angles, in other words with 0˚ knee flexion the CV was higher compared to 90˚ knee flexion (Table 4). In addition, for some participants the CV was above 3%. This percentage also dropped with increasing knee flexion. Intra-operator reliability did not differ significantly between dominant and non-dominant knees (p = 0.493 for 0˚, p = 0.889 for 45˚, and p = 0.831 for 90˚ knee flexion). CV did not differ significantly between sexes or over age (S2 Table).

**Table 4. Intra-operator reliability*.**

| Knee | Angle of knee flexion | Stiffness (N/m) | CV (%) | p-value | CV above 3% border, amount (%) |
|------|----------------------|-----------------|--------|---------|-------------------------------|
| Dominant | 0˚ | 351.5±123.2 | 2.4±1.8 | <0.001 | 12 (27%) |
| | 45˚ | 681.5±126.3 | 1.7±1.2 | | 4 (9%) |
| | 90˚ | 834.6±153.8 | 1.2±0.8 | | 2 (4%) |
| Non-dominant | 0˚ | 327.8±92.9 | 2.1±1.5 | 0.001 | 10 (22%) |
| | 45˚ | 665.1±124.5 | 1.6±0.92 | | 5 (11%) |
| | 90˚ | 833.3±125.3 | 1.2±0.8 | | 1 (2%) |

*Based on three measurements performed per knee angle, in 45 volleyball players.

Average stiffness ± standard deviation and coefficient of variation (CV) of the patellar tendon are shown. The p-values represent the difference in CV between knee angles.

**Table 5. Measurement error results\*.**

| Knee | Knee flexion angle | SEM (N/m) | MDC (N/m) | 95% LOA | | 95%CI mean diff | |
|---|---|---|---|---|---|---|---|
| | | | | Lower | Upper | Lower | Upper |
| Dominant | 0˚ | 13.5 | 37.3 | -37.5 | 37.9 | -5.6 | 6.0 |
| | 45˚ | 15.4 | 42.8 | -45.4 | 42.1 | -8.4 | 5.0 |
| | 90˚ | 14.5 | 40.3 | -40.6 | 39.6 | -6.6 | 5.6 |
| Non-dominant | 0˚ | 11.0 | 30.4 | -28.4 | 32.3 | -2.7 | 6.6 |
| | 45˚ | 15.2 | 42.1 | -44.9 | 37.8 | -9.9 | 2.8 |
| | 90˚ | 14.3 | 39.5 | -41.3 | 37.7 | -7.9 | 4.3 |

\*Based on three measurements performed per knee angle, in 45 volleyball players.

Abbreviations: 95%CI mean diff, 95% confidence interval of the mean difference; 95% LOA, 95% limits of agreement; MDC, minimal detectable change; SEM, standard error of measurement.

SEM and MDC values did not differ between knee angles (Table 5). The 95% LOA values were wider for the dominant leg compared to the non-dominant leg, however for SEM and MDC no clear distinction could be made between knee angles.

## Discussion

This study aimed to evaluate intra- and inter-operator reliability of a handheld digital palpation device in measuring compressive stiffness of the patellar tendon at different knee angles in talent and elite volleyball players. Second aim was to examine differences in reliability when measuring at different knee angles, between dominant and non-dominant knees, between sexes, and with age. For intra-operator reliability, the Myoton appeared to be a reliable device to measure stiffness among talent and elite volleyball players. For inter-operator reliability, the measured stiffness differed significantly between two operators. This might challenge repetitive use of tendotonometry by different operators.

No differences in reliability could be found when measuring dominant and non-dominant knees, different sexes, and different ages. However, reliability significantly improved with increasing knee flexion angle. One should keep in mind that stiffness rises with knee angle, there is no difference between dominant and non-dominant knees, and men have a higher patellar tendon stiffness than to women.

**Table 6. Inter-operator reliability\*.**

| Participant | Stiffness operator 1 (N/m) | CV's of operator 1 (%) | Stiffness operator 2 (N/m) | CV's of operator 2 (%) | Difference in stiffness between operators (N/m) | p-value |
|---|---|---|---|---|---|---|
| 1 | 514.7±18.4 | 3.6 | 458.7±15.6 | 3.4 | 55.9±8.7 | <0.001 |
| 2 | 348.8±8.3 | 2.4 | 524.7±17.6 | 3.4 | 175.9±10.0 | <0.001 |
| 3 | 929.7±13.8 | 1.5 | 947.7±12.5 | 1.3 | 17.9±4.5 | <0.001 |
| 4 | 578.7±11.1 | 1.9 | 615.7±20.5 | 2.3 | 37.1±11.2 | <0.001 |
| 5 | 780.4±8.3 | 0.9 | 749.1±36.6 | 4.9 | 33.8±26.6 | 0.007 |
| **Average** | 628.2±204.5 | 2.0 | 659.0±176.6 | 3.1 | 64.1±59.0 | |

\*Based on 15 measurements performed by two operators, in 5 volleyball players.

Average stiffness ± standard deviation and Coefficient of Variation (CV) of patellar tendon are shown for the 15 consecutive stiffness measurements performed per participant.

## Intra-operator reliability

With excellent ICC results and CVs below 3%, the Myoton can be considered a reliable device to measure stiffness of the patellar tendon among talent and elite volleyball players when measurements are performed by one operator. Only two articles have investigated reliability of stiffness measurements [20,24]; both studies only performed limited statistical testing, without CV [20,24] or only looking at the ICC. Based on the ICC one can easily conclude that a device is reliable, yet it is more precise to draw conclusions based on the CV [24]. Both articles investigated reliability in a different population (inactive [20] and recreationally active [24]) compared to the talent and elite volleyball population in the current study. Albeit with moderate (for Achilles tendon) and inconclusive (for PT) evidence, both studies found good intra-operator reliability of the Myoton in measuring stiffness. These results are in line with our findings.

Noteworthy, we measured intra-operator reliability while keeping the marker visible in-between measurements, which allowed us to measure intra-operator reliability of the device itself. In practice, the marker will be removed in-between measurements, as will participants' positioning. Future research should therefore focus on determining intra-operator reliability of the *measurement procedure* with the Myoton, which will enrich the current knowledge about the reliability of the device itself. The ICC values are probably lower in that setting.

## Inter-operator reliability

Significant differences were found in stiffness when measured by two different operators. Although the ICC resulted in excellent reliability, these significant differences in stiffness raise the question of whether tendotonometry is reliable when used by different operators. A possible explanation for these differences might be the different positioning of the Myoton. After the measurements of the first operator, the marking point at the skin was removed and thereafter the second operator had to determine the correct location for the measurements again. It therefore remains unknown whether the difference in stiffness is due to the different placement on the tendon on which stiffness was measured, or whether it is really due to the other operator's measurement. Nevertheless, the way we measured differences between two operators is common in daily practice. We recommend focusing on proper placement of the Myoton by a trained examiner who has proper anatomical knowledge.

Only one other paper was found investigating inter-operator reliability of the Myoton measuring patellar tendon stiffness [20]; it also found excellent ICC values ranging between 0.78 and 0.98. Although that study describes the locations of measurement in great detail, nothing is mentioned about removing or keeping the marking points in-between measurements within and between operators.

Additional studies were found investigating inter-operator reliability of the Myoton in measuring stiffness of the Achilles tendon [19,21–23], yielding excellent inter-operator reliability, all with ICC values above 0.76. However, none reported absolute differences in stiffness values as measured by different operators. We also found high ICC values in our current study, but with significant differences between operators. One might therefore question the findings of these previous studies.

In addition, none of those studies mention anything about removal of the markers in-between measurements. Two of the studies only mention that markers were placed by a physical therapist who did not take the measurements [19,21]. It can therefore be assumed that there was no removal of markers in-between measurements within and between operators. That implies that the design of those studies is similar to ours, and results are comparable (although determined in another tendon).

## Factors influencing reliability

The occurrence of CV's above 3% significantly rose with decreasing knee angle. This suggests that reliability improves with increasing knee angles. Only one similar study measured stiffness at different knee angles [20], also finding increased reliability with increasing angles, with optimal reliability at 90° knee flexion [20]. This change in reliability might be due to the subcutaneous fat around the joint [28,29]. Subcutaneous fat is more prominent at the joint with an extended knee. This might influence measurements of a slack PT. When the knee is flexed to 90° the tendon remains in a strained status, where the subcutaneous fat has less influence on the measurement. This suggests that the best knee angle to measure stiffness is 90°.

As we found no significant differences between dominant and non-dominant knee, females and males, or age when it comes to the reliability of tendotonometry, these factors may not be influential. No other studies could be found that investigated these factors.

## Other methods to measure stiffness

Next to "tendotonometry" there are several other ways to measure stiffness of tendon tissue, for example, using magnetic resonance elastography (MRE) [30–32], or strain elastography [33,34], or calculating by dividing force by deformation [35–37]. However, to our knowledge nothing is known about the reliability of these methods in measuring tendon stiffness or reliability is questionable [33,38,39].

Another, frequently used method to measure stiffness of tendon tissue is shear wave elastography (SWE). Although it is known to be rather expensive, requiring ultrasound equipment and experienced operators, it is also rather reliable in measuring stiffness of muscle and tendon tissue [14,40,41]. Intra- and inter-operator reliability of SWE are comparable to the currently investigated Myoton, with the same advice to use a single operator. However, SWE has a higher variance in equipment, analyses, shear wave velocities with different transducers and different acquisition depths, and joint positioning [14].

Another version of the SWE technique is continuous shear wave elastography (cSWE), which is a modification to the supersonic shear imaging (SSI) technique [42]. cSWE has been found to be reliable and valid in measuring stiffness in healthy Achilles tendons as well as pathological patellar tendons [41,43]. Still, in addition to MRE and SWE many variables influence the outcome and expensive equipment is needed with skilled operators, making these techniques less practicable for use in regular and sports practice.

Tendon stiffness can also be determined using a dynamometer and B-mode ultrasonography recordings (DBUS). Although this method has been found to be valid and reliable, it can be challenging since participants have to perform a maximal voluntary contraction [33,44–47].

## Strengths and limitations

A strength of the current study is that we performed the measurements among elite volleyball players. It is crucial to investigate reliability of tendotonometry in a population with a high prevalence of PT injuries [1,48]. Although performing measurements in elite athletes can be challenging, we accomplished this in a large group of elite volleyball players during their normal daily training routines. In the future, tendotonometry might be used to monitor compressive stiffness and link it to tendon injuries in this specific population. A second strength is that we used a wide array of statistical tests investigating different aspects of reliability. A third strength is the way we measured inter-operator reliability. Although the marker was removed in-between operators, which makes it difficult to measure the reliability of the device itself, this is the way it also will be performed in practice, hence the inter-operator reliability found in this study is applicable to daily practice. A fourth and final strength is that our operators were

blinded to their own results and those of others. This increases the methodological quality of the current study.

A limitation of our study is that we measured stiffness at 0° and 45° knee flexion with feet placed on the bench, while stiffness at 90° knee flexion was measured with feet hovering off the floor. This could have influenced the stiffness of the PT and also the reliability measurements at these different angles. A knee angle of 90° can be standardized more easily when feet are placed on the bench, lying supine, since the knee angle sitting on the bench is largely influenced by the size of the upper leg. The larger the cross-sectional area of the upper leg, the smaller the knee angle becomes, lowering reliability as found in the current study. In a future study positioning of participants might be investigated, having them seated as well as lying on the bench, holding their knees in 90° (with and without hip flexion). This would allow a clear conclusion to be drawn on the influence of different joint (hip, ankle) positions on compressive stiffness. A second limitation is that muscle activity was not measured during stiffness measurements. Although participants were instructed to remain fully relaxed, changes in stiffness may occur with changes in muscle activity [49]. In future studies EMG measurements may help show complete muscle relaxation during tendotonometry [50,51]. Third, little is known about the validity of the Myoton in measuring stiffness: we found only one study investigating its validity, comparing it with SWE [16], so more research is needed. One final note of attention is that stiffness was measured perpendicular to the PT, while the actual tendon stiffness runs longitudinal.

## Recommendations

*For practical use:*

- Measure stiffness with a knee angle of 90° flexion, fixating the foot on the treatment table and instructing subjects to relax their muscles as much as possible.

- The handheld device gives a CV value after every measurement. When this CV is above 3%, re-measure.

*Future studies should investigate*:

- The influence of different positioning strategies with respect to ankle and hip angles on the compressive stiffness.

- Intra-operator reliability including all human activities, which means removing the marker on the skin. In such a study design, the operator places the mark on the skin again each time and removes it in-between measurements. The operator puts the participant in the correct position again each time and measures as if the participant was entering for a new data collection bout.

- The position of the probe on the tendon, with measurements closer to the patellar apex and measurements closer to the tibial tuberosity.

## Conclusion

For intra-operator reliability, the Myoton appeared to be a reliable device to measure compressive stiffness in Dutch young elite volleyball players, especially at a knee angle of 90° flexion measured by a single operator. However, inter-operator reliability appeared questionable. In order to improve reliability, standardized positioning seems necessary.

## Supporting information

**S1 Fig. The Bland-Altman plots.** Upper row: dominant knee, lower row: non-dominant knee. Since measurements were performed in triplicate and with Bland-Altman plots only two can be compared; the columns from left to right represent the comparisons between measurements 1 and 2, measurements 1 and 3, and measurements 2 and 3. In each figure, the bold black line represents the mean of the difference, the small dotted lines represent the upper and lower limits of agreement (LOA), and the long dotted lines the 95% confidence interval of the mean difference. B.1: 0˚ knee flexion, B.2: 45˚ knee flexion, B.3: 9˚ knee flexion.
(TIF)

**S1 Table. Compressive stiffness.**
(DOCX)

**S2 Table. P-values of the differences in reliability.**
(DOCX)

**S1 Dataset.**
(SAV)

**S2 Dataset.**
(XLSX)

## Acknowledgments

We thank Caroline Roozenboom-van Vliet and Mireille A Baart (Gelderse Vallei Hospital, Ede, The Netherlands) for their advice on the statistical analyses. We thank Ruth Rose for improving the language of the article.

## Author Contributions

**Conceptualization:** Lotte van Dam, Rieneke Terink, Inge van den Akker-Scheek, Johannes Zwerver.

**Data curation:** Lotte van Dam, Rieneke Terink, Inge van den Akker-Scheek, Johannes Zwerver.

**Formal analysis:** Lotte van Dam, Rieneke Terink, Inge van den Akker-Scheek, Johannes Zwerver.

**Funding acquisition:** Lotte van Dam, Rieneke Terink, Inge van den Akker-Scheek, Johannes Zwerver.

**Investigation:** Lotte van Dam, Rieneke Terink, Inge van den Akker-Scheek, Johannes Zwerver.

**Methodology:** Lotte van Dam, Rieneke Terink, Inge van den Akker-Scheek, Johannes Zwerver.

**Project administration:** Lotte van Dam, Rieneke Terink, Inge van den Akker-Scheek, Johannes Zwerver.

**Resources:** Lotte van Dam, Rieneke Terink, Inge van den Akker-Scheek, Johannes Zwerver.

**Software:** Lotte van Dam, Rieneke Terink, Inge van den Akker-Scheek, Johannes Zwerver.

**Supervision:** Lotte van Dam, Rieneke Terink, Inge van den Akker-Scheek, Johannes Zwerver.

**Validation:** Lotte van Dam, Rieneke Terink, Inge van den Akker-Scheek, Johannes Zwerver.

**Visualization:** Lotte van Dam, Rieneke Terink, Inge van den Akker-Scheek, Johannes Zwerver.

**Writing – original draft:** Lotte van Dam, Rieneke Terink, Inge van den Akker-Scheek, Johannes Zwerver.

**Writing – review & editing:** Lotte van Dam, Rieneke Terink, Inge van den Akker-Scheek, Johannes Zwerver.

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
