## [Decision Letter · Decision Letter 0]

21 Feb 2024

PONE-D-24-01254Intra- and inter-operator reliability of measuring patellar tendon stiffness using tendotonometry in volleyball playersPLOS ONE

Dear Dr. van Dam,

Thank you for submitting your manuscript to PLOS ONE. After careful consideration, we feel that it has merit but does not fully meet PLOS ONE’s publication criteria as it currently stands. Therefore, we invite you to submit a revised version of the manuscript that addresses the points raised during the review process.

We look forward to receiving your revised manuscript.

Kind regards,

Charlie M. Waugh

Academic Editor

PLOS ONE

“This project was funded by a SportInnovator Voucher from ZonMw (file number: 538001779)”

3. In this instance it seems there may be acceptable restrictions in place that prevent the public sharing of your minimal data. However, in line with our goal of ensuring long-term data availability to all interested researchers, PLOS’ Data Policy states that authors cannot be the sole named individuals responsible for ensuring data access (http://journals.plos.org/plosone/s/data-availability#loc-acceptable-data-sharing-methods).

4. Please amend the manuscript submission data (via Edit Submission) to include author Hans Zwerver.

5. Please amend your authorship list in your manuscript file to include author Johannes Zwerver.

6. We notice that your supplementary figure (Figure S2) is uploaded with the file type 'Figure'. Please amend the file type to 'Supporting Information'. Please ensure that each Supporting Information file has a legend listed in the manuscript after the references list.

Reviewers' comments:

Reviewer's Responses to Questions

**Comments to the Author**

1. Is the manuscript technically sound, and do the data support the conclusions?

Reviewer #1: Partly

Reviewer #2: Yes

2. Has the statistical analysis been performed appropriately and rigorously? 

Reviewer #1: Yes

Reviewer #2: Yes

3. Have the authors made all data underlying the findings in their manuscript fully available?

Reviewer #1: No

Reviewer #2: Yes

4. Is the manuscript presented in an intelligible fashion and written in standard English?

Reviewer #1: No

Reviewer #2: Yes

5. Review Comments to the Author

Reviewer #1: Lotte van Dam and colleagues examine both intra- and inter-operator reliability of the MyotonPRO in volleyball players. This device has several applications in the research and clinical world – determining its reliability for assessing tendon biomechanical properties is highly applicable. While the topic was interesting, there are concerns about the protocol for assessing intra-operator reliability of the device.

Major comments

• Line 102, the author states that the midpoint of the PT is marked by one trained operator and the location remained visible throughout measurements (for intra-operator assessment). As tendon composition is not homogeneous throughout its entire length, measurement location would affect MyotonPRO outcomes (as the author postulated on line 238 in relation to inter-observer differences). Therefore, determining the measurement location on the tendon is an integral part of collecting the biomechanical properties. Typically, determining intra-operator reliability demands that all marks, positions, and measurements would be made as if the participant was entering for a new data collection bout. In the current form (using the same pre-marked measurement location) the ICC values are likely inflated - presenting severe uncertainty regarding the reported intra-observer reliability values.

• Although this manuscript is poised to provide valuable insight for researchers and clinicians, the writing is often non-scientific and verbose, leading to areas that lacked clarity. Further, the author needs to make another run through for basic edits.

• Traditionally, tendon stiffness refers to longitudinally oriented properties of the tendon (tensile). The MyotonPRO loads the tendon perpendicularly - the relationship between this outcome and tensile stiffness is not currently clear. Therefore, is it appropriate to use the term ‘tendon stiffness’? One of your references (Finnamore et al.) uses the term transverse stiffness, perhaps it would be appropriate to use this distinction.

• I’m curious about the usage of ‘tendotonometer’, a term that does not show up in any of the literature. The manufacturers describe the MyotonPRO as a hand-held digital palpation device. I’ve also seen it described as a ‘hand-held dynamometer’.

• The discussion needs to be broadened - the author should reinforce by giving greater detail and providing more references to the literature.

Minor comments

• Abstract - please add study design

• Line 56 – For justification, more information needed about limitations in the referenced studies. Noted that there is a statement on line 223.

• Line 79 – Please add more information for musculoskeletal dysfunction. Is this all prior injuries, current injury, just to the PT or lower limbs?

• Line 92 – Please provide more information about the device settings/parameters, e.g. impulse force, impulse time, pre-loading…

• Line 99-100 – “PT stiffness was measured halfway between the patellar apex and the tibial tuberosity”. Perhaps consolidate into ‘Tendonometer’ section.

• Line 104 – Make and model for goniometer.

• Line 105-107 – Please give more participant positioning information. How knee angle was maintained in 45 degree measurements, etc.

• Line 117 – change ‘intra-operator’ for ‘inter-operator’

• Line 118 – Authors need to be clearer about the number of measurements/impulses. It can be confusing when jumping between total impulses administered and measurement bouts.

• Table 1.

o Change ‘Length’ to ‘Height’.

o Dominant leg, 41 left to 4 right

o Stiffness – n/mm

• Line 165 – As this is one of your main outcomes, it should be presented in the manuscript text or tables. Further, all ICC/CI values were >0.97, which is much more reliable than ‘above 0.9’.

• Line 224-225 – Statement and argument needs to be clearer and justified. The device has an internal measurement of CV per trial (5 impulses), reference (Finnamore et al) state that they re-administered trials if CV >3%.

• Line 254-255 – This is an extremely definitive sentence; softer language would be appropriate when it comes to conclusions drawn.

• Line 260 – Needs citation

Reviewer #2: General comments:

The paper is generally well written and provides relevant information in the field of tendon clinical biomechanics. However, key areas for improvement and clarification are needed.

The term "tendotonometry" has not been used before. The Myoton gives several mechanical variables derived from the compressive mechanical analysis that depend on several tissues such as skin, subcutaneous fat tissue, and fascia. Therefore, using the term "compressive stiffness" is more appropriate than "tendomyometry", considering that it is not possible to isolate the compressive mechanical properties of the tendon. That is why some authors refer to the term "compressive stiffness."

In addition, a better explanation is needed on the argumentation of measuring the patellar tendon compressive stiffness using different knee joint angles. For example, in the Achilles tendon, the stiffness can change depending on the knee and ankle joint position. Furthermore, a better justification of Myoton is needed compared to other passive techniques, such as shear wave ultrasound or dynamic measurement, such as ultrasound assessment of tendon displacement during isometric contraction.

Considering that the authors have nice data, I suggest including the interaction between joint angle and sex (Mixed ANOVA). The authors already have the sex comparison in Table 1 but only use the 90-degree knee joint angle. This comparison may help us better understand the results and help future studies use the data as comparative values, not only for reliability.

Specific comments:

Introduction:

The authors also need to clarify the advantage of measuring with the Myoton, specifically noting its comprehensive nature involving skin, adipose tissue, and fascia, in addition to its advantage in contrast to other techniques.

Method:

Does the patellar tendon compressive stiffness also differ in the proximal, middle, and distal portions? Please clarify where the tendon was measured and justify why this portion was selected.

Measuring different knee joint angles to see the changes in tendon compressive stiffness needs to ensure that individuals are relaxed without muscle activity. The EMG activity may help to ensure muscle relaxation. Please include it as a limitation.

Additionally, more background information on subjects, such as history of surgery or tendinopathy, is needed.

Results:

Table 1 only includes the values of stiffness at 90 degrees of the knee joint. Are there significant differences between angles?

Strength of study:

It would be nice if the authors included a sentence with recommendations for future research and clinical/sports practice, such as recommendations about the positions of the knee, hip, and foot.

Limitation:

Why do the authors suggest that feet on the floor can be a limitation? Does the cutaneous stimulus trigger muscle activity and tone of the quadriceps muscles? Please clarify.

6. PLOS authors have the option to publish the peer review history of their article (what does this mean?). If published, this will include your full peer review and any attached files.

Reviewer #1: No

Reviewer #2: No

---

## [Author Response · Author response to Decision Letter 0]

5 Apr 2024

Response to editor and referees of manuscript PONE-D-24-01254

Intra- and inter-operator reliability of measuring patellar tendon stiffness using tendotonometry in volleyball players

Dear editor,

Thank you for offering the opportunity to revise the manuscript entitled “Intra- and inter-operator reliability of measuring patellar tendon stiffness using tendotonometry in volleyball players” (PONE-D-24-01254). We are thankful to the reviewers for their useful and constructive feedback, which has further improved our manuscript. 

Below you will find our reply to the comments, which have been incorporated into the revised manuscript. We are submitting the revised manuscript using tracked changes.

Thank you for your time and effort to evaluate our revised manuscript. 

We look forward to your decision.

Yours sincerely,

Also on behalf of the co-authors,

Lotte van Dam, MSc

Reviewer 1:

Lotte van Dam and colleagues examine both intra- and inter-operator reliability of the MyotonPRO in volleyball players. This device has several applications in the research and clinical world – determining its reliability for assessing tendon biomechanical properties is highly applicable. While the topic was interesting, there are concerns about the protocol for assessing intra-operator reliability of the device.

Major comments

Comment 1:

Line 102, the author states that the midpoint of the PT is marked by one trained operator and the location remained visible throughout measurements (for intra-operator assessment). As tendon composition is not homogeneous throughout its entire length, measurement location would affect MyotonPRO outcomes (as the author postulated on line 238 in relation to inter-observer differences). Therefore, determining the measurement location on the tendon is an integral part of collecting the biomechanical properties. Typically, determining intra-operator reliability demands that all marks, positions, and measurements would be made as if the participant was entering for a new data collection bout. In the current form (using the same pre-marked measurement location) the ICC values are likely inflated - presenting severe uncertainty regarding the reported intra-observer reliability values.

Answer 1:

We agree with the reviewer that there is a clear distinction in intra-operator reliability when keeping and removing the marker in-between measurements. When keeping the marker, the variability in results is purely because of the device. When removing the marker in-between measurements one does not know whether the variability came from variability in the device or from measuring at a different place on the tendon. Since we wanted to know the pure variability of the device itself, we chose the current method. To be more clear on this topic, we have tried to describe it throughout the whole manuscript by saying “reliability of the device”, for example: “This study aimed to evaluate intra- and inter-operator reliability of a handheld digital palpation device in measuring compressive transverse stiffness of the patellar tendon at different knee angles in talent and elite volleyball players.” in lines 59-62, “Two different operators measured stiffness to determine inter-operator reliability of the Myoton.” in line 126 and “Noteworthy, we measured intra- operator reliability while keeping the marker visible in-between measurements, which allowed us to measure intra- operator reliability of the device itself. In practice, the marker will be removed in-between measurements, as will participants’ positioning. Future research should therefore focus on determining intra-operator reliability of the measurement procedure with the Myoton, which will enrich the current knowledge about the reliability of the device itself. The ICC values are probably lower in that setting.” in lines 244-249.

Comment 2:

Although this manuscript is poised to provide valuable insight for researchers and clinicians, the writing is often non-scientific and verbose, leading to areas that lacked clarity. Further, the author needs to make another run through for basic edits.

Answer 2:

We thank the reviewer for this feedback to improve the manuscript with more scientific and less verbose writing. We critically ran through the manuscript to change the writing where we considered it necessary, and also based on the suggestions of both reviewers. We hope these adaptations are more in line with the reviewer’s expectations. Additionally, we asked an English-language professional to copyedit the manuscript. 

Comment 3:

Traditionally, tendon stiffness refers to longitudinally oriented properties of the tendon (tensile). The MyotonPRO loads the tendon perpendicularly - the relationship between this outcome and tensile stiffness is not currently clear. Therefore, is it appropriate to use the term ‘tendon stiffness’? One of your references (Finnamore et al.) uses the term transverse stiffness, perhaps it would be appropriate to use this distinction.

Answer 3: 

We agree with the reviewer that the actual tendon stiffness is longitudinal stiffness and that we measured perpendicularly, in a transverse plane. Therefore we changed ‘tendon stiffness’ into ‘compressive transverse stiffness’ or ‘stiffness’ throughout the whole manuscript. We also changed this in the title of the manuscript: “Intra- and inter-operator reliability of measuring compressive transverse stiffness of the patellar tendon in volleyball players using a handheld digital palpation device”. We have also described the method of measuring perpendicularly in the Method section, for example “). The Myoton is a non-invasive handheld digital palpation device for compressive and transverse measuring of muscle, tendon, and other soft tissue properties.” in lines 85-87, and “The probe of the device was held perpendicular to the skin overlying the PT” in lines 92-93. 

In addition, we wrote a small section about transverse and longitudinal stiffness in the limitations section: “One final note of attention is that stiffness was measured perpendicular to the PT, while the actual tendon stiffness runs longitudinal.”, see lines 343-344.

Comment 4:

I’m curious about the usage of ‘tendotonometer’, a term that does not show up in any of the literature. The manufacturers describe the MyotonPRO as a hand-held digital palpation device. I’ve also seen it described as a ‘hand-held dynamometer’.

Answer 4:

The term ‘tendotonometer’ was invented for this manuscript for the device (MyotonPRO) that measures transverse stiffness of tendon tissue. By the word ‘Myoton’ and ‘myotonometer’ (also seen in literature) one could assume that we are measuring properties of the muscle. We thought this was unsuitable since we measure tendon tissue. However, we agree with the reviewer that the terminology might be confusing and we now have changed the ‘tendotonometer’ into the original name of the device, ‘Myoton’. For the method of measuring transverse stiffness of tendon tissue we keep the self-invented term ‘tendotonometry’. We think this word encompasses the measurement method very well and it is in line with terminology used in literature to describe the muscle tissue measuring method (myotonometry). If the reviewer does not agree with this reasoning, we are flexible.

Comment 5:

The discussion needs to be broadened - the author should reinforce by giving greater detail and providing more references to the literature.

Answer 5: 

Thank you for this feedback. We have added greater detail where possible, plus provided more references to the literature where needed, for example in the part about inter-operator reliability: “Only one other paper was found investigating inter-operator reliability of the Myoton measuring patellar tendon stiffness [20]; it also found excellent ICC values ranging between 0.78 and 0.98. Although that study describes the locations of measurement in great detail, nothing is mentioned about removing or keeping the marking points in-between measurements within and between operators.

Additional studies were found investigating inter-operator reliability of the Myoton in measuring stiffness of the Achilles tendon [19, 21-23], yielding excellent inter-operator reliability, all with ICC values above 0.76. However, none reported absolute differences in stiffness values as measured by different operators. We also found high ICC values in our current study, but with significant differences between operators. One might therefore question the findings of these previous studies. 

In addition, none of those studies mention anything about removal of the markers in-between measurements. Two of the studies only mention that markers were placed by a physical therapist who did not take the measurements [19, 21]. It can therefore be assumed that there was no removal of markers in-between measurements within and between operators. That implies that the design of those studies is similar to ours, and results are comparable (although determined in another tendon).”, lines 262-278.

In addition, we have added some more information about other stiffness measurement techniques in addition to the Myoton in a separate sub-section: “Next to “tendotonometry” there are several other ways to measure stiffness of tendon tissue, for example, using magnetic resonance elastography (MRE) [30-32], or strain elastography [33, 34], or calculating by dividing force by deformation [35-37]. However, to our knowledge nothing is known about the reliability of these methods in measuring tendon stiffness or reliability is questionable [33, 38, 39].

Another, frequently used method to measure stiffness of tendon tissue is shear wave elastography (SWE). Although it is known to be rather expensive, requiring ultrasound equipment and experienced operators, it is also rather reliable in measuring stiffness of muscle and tendon tissue [14, 40, 41]. Intra- and inter-operator reliability of SWE are comparable to the currently investigated Myoton, with the same advice to use a single operator. However, SWE has a higher variance in equipment, analyses, shear wave velocities with different transducers and different acquisition depths, and joint positioning [14]. 

Another version of the SWE technique is continuous shear wave elastography (cSWE), which is a modification to the supersonic shear imaging (SSI) technique [42]. cSWE has been found to be reliable and valid in measuring stiffness in healthy Achilles tendons as well as pathological patellar tendons [41, 43]. Still, in addition to MRE and SWE many variables influence the outcome and expensive equipment is needed with skilled operators, making these techniques less practicable for use in regular and sports practice.

Tendon stiffness can also be determined using a dynamometer and B-mode ultrasonography recordings (DBUS). Although this method has been found to be valid and reliable, it can be challenging since participants have to perform a maximal voluntary contraction [33, 44-47].”, lines 292-313.

Minor comments

6. Abstract - please add study design

Answer 6: 

We added the ‘observational’ study design in the abstract, “This observational study aimed to evaluate…”, see line 20.

7. Line 56 – For justification, more information needed about limitations in the referenced studies. Noted that there is a statement on line 223.

Answer 7: 

We have added statistical testing as a limitation in the referenced studies: “We found only two articles investigating the reliability of the Myoton, both with methodological limitations, such as statistical tests performed.” see lines 57-59.

8. Line 79 – Please add more information for musculoskeletal dysfunction. Is this all prior injuries, current injury, just to the PT or lower limbs?

Answer 8: 

We have added that participants were excluded if they had any current musculoskeletal dysfunction, “Participants were excluded if they had any current musculoskeletal dysfunction or took medication that could affect musculoskeletal function.”, see lines 80-81. 

9. Line 92 – Please provide more information about the device settings/parameters, e.g. impulse force, impulse time, pre-loading…

Answer 9: 

We have added more information about the device settings and parameters, such as deformation area, pre-compression strength and impulse force, “The probe of the device was held perpendicular to the skin overlying the PT (±5°, monitored by the device itself), with a deformation area of 7.1mm2. Next, the probe was pushed against the skin to reach the correct depth. This was signaled by a red light turning green on the Myoton, indicating a pre-compression strength of 0.18N. Then, five short impulses of 0.4N and with a tap interval of 0.8sec, were automatically applied by the device, to induce mechanical, damped oscillations in the underlying tissues.”, see lines 92-98.

10. Line 99-100 – “PT stiffness was measured halfway between the patellar apex and the tibial tuberosity”. Perhaps consolidate into ‘Tendonometer’ section.

Answer 10: 

In the ‘tendotonometry’ section we wanted to focus on the technical and functional aspects of the device, how the device works and what it measures. In the ‘study procedure’ section we wanted to be more precise about how we have used the device, in which angles, and in which position we measured on the tendon, to clearly distinguish between technical aspects and practical usage. However, we understand the reviewers confusion and to make this distinction more clear we changed the subheading into ‘study procedure’. 

11. Line 104 – Make and model for goniometer.

Answer 11: 

We have added the brand, and model of the digital goniometer to the manuscript, with a reference of its validation: “A calibrated and validated digital goniometer (Goniometer Pro, Android version, Digiflex Labs, Seattle, UAS) [24], was used to quantify the angle of knee flexion.”, see lines 110-112.

12. Line 105-107 – Please give more participant positioning information. How knee angle was maintained in 45 degree measurements, etc.

Answer 12: 

Thank you for pointing this out, it was indeed missing. Some more information is now given in the revised manuscript about positioning of the participant, including how the knee angle was maintained in 45º measurements: “For the 90º knee angle, the participant sat in upright position with feet hovering off the floor. At 0º and 45º knee flexion, the participant was in supine position with legs extended (0º) or passively half-flexed (45º) and supported by the examination table (Fig 1). Participants were instructed to lie fully relaxed.” see lines 112-115.

13. Line 117 – change ‘intra-operator’ for ‘inter-operator’

Answer 13: 

Thank you for noticing, we have changed this typo. 

14. Line 118 – Authors need to be clearer about the number of measurements/impulses. It can be confusing when jumping between total impulses administered and measurement bouts.

Answer 14: 

We agree that this may lead to confusion. For clarity’s sake we have created an additional table presenting how many measurements were performed for intra- and inter-operator reliability, see Table 1 on page 6 in the manuscript. 

Table 1. Overview of measurements performed to determine reliability of the Myoton.

 Intra-operator reliability Inter-operator reliability

No. operators 1 2

No. participants 45 5

No. knee angles 3 (0°, 45°, 90°) 1 (90°)

No. measurements per knee angle 3 15

15. Table 1.

• Change ‘Length’ to ‘Height’.

• Dominant leg, 41 left to 4 right

• Stiffness – n/mm

Answer 15: 

The mentioned details were added or changed to Table 1.

16. Line 165 – As this is one of your main outcomes, it should be presented in the manuscript text or tables. Further, all ICC/CI values were >0.97, which is much more reliable than ‘above 0.9’.

Answer 16: 

Since we agree with the reviewer, we have added the table from the supporting information to the manuscript itself, as Table 3 on page 9. In addition, we changed the 0.9 into 0.97 in the text: “All ICC values including the confidence intervals were above 0.970 (Table 3)” (line 179).

17. Line 224-225 – Statement and argument needs to be clearer and justified. The device has an internal measurement of CV per trial (5 impulses), reference (Finnamore et al) state that they re-administered trials if CV >3%.

---

## [Decision Letter · Decision Letter 1]

17 May 2024

Intra- and inter-operator reliability of measuring compressive transverse stiffness of the patellar tendon in volleyball players using a handheld digital palpation device

PONE-D-24-01254R1

Dear Dr. van Dam,

We’re pleased to inform you that your manuscript has been judged scientifically suitable for publication and will be formally accepted for publication once it meets all outstanding technical requirements.

Kind regards,

Charlie M. Waugh

Academic Editor

PLOS ONE

Additional Editor Comments (optional):

Reviewer 1 still believes that the wording of 'intra-operator reliability' could be made clearer, specifically that this test is being performed just on the device and limits the applicability of the results to any form of clinical or research based use.

Reviewer 2 believes the term "compressive transverse stiffness" is confusing, and recommends to change it to "compressive stiffness" throughout the manuscript (including the title).

Since these are minor comments that do not change the objectives or conclusions of the paper, I am choosing to accept the manuscript in its current form, but urge you to consider these comments when editing your proofs.

Reviewers' comments:

Reviewer's Responses to Questions

**Comments to the Author**

1. If the authors have adequately addressed your comments raised in a previous round of review and you feel that this manuscript is now acceptable for publication, you may indicate that here to bypass the “Comments to the Author” section, enter your conflict of interest statement in the “Confidential to Editor” section, and submit your "Accept" recommendation.

Reviewer #1: All comments have been addressed

Reviewer #2: All comments have been addressed

2. Is the manuscript technically sound, and do the data support the conclusions?

Reviewer #1: Yes

Reviewer #2: Yes

3. Has the statistical analysis been performed appropriately and rigorously? 

Reviewer #1: Yes

Reviewer #2: Yes

4. Have the authors made all data underlying the findings in their manuscript fully available?

Reviewer #1: Yes

Reviewer #2: Yes

5. Is the manuscript presented in an intelligible fashion and written in standard English?

Reviewer #1: Yes

Reviewer #2: Yes

6. Review Comments to the Author

Reviewer #1: (No Response)

Reviewer #2: The authors have included all the suggestions. However, the term "compressive transverse stiffness" is confusing. It is relatively well documented that the mechanism of myotonPro is based on compressive stiffness; however, the "transverse" effect is difficult to introduce without specific documentation or data to support this fact. I suggest simplifying the term to "compressive stiffness" and validating the term with more literature that has used it before. I recommend changing the term "compressive transverse stiffness" to "compressive stiffness" throughout the article, including the title. This way, the paper can be easier to understand, motivate readers, and encourage citations.

7. PLOS authors have the option to publish the peer review history of their article (what does this mean?). If published, this will include your full peer review and any attached files.

Reviewer #1: No

Reviewer #2: No

---

## [Editor Report · Acceptance letter]

30 May 2024

PONE-D-24-01254R1 

PLOS ONE

Dear Dr. van Dam, 

I'm pleased to inform you that your manuscript has been deemed suitable for publication in PLOS ONE. Congratulations! Your manuscript is now being handed over to our production team.

Kind regards, 

on behalf of

Dr. Charlie M. Waugh 

Academic Editor

PLOS ONE